# OpenReview forum: "SyMerge: From Non-Interference to Synergistic Merging via Single-Layer Adaptation"
_ICML.cc/2026/Conference — ICML 2026 regular_

### Official Review · Reviewer_MNZH · 2026-03-03

**Soundness:** 3
**Presentation:** 3
**Significance:** 2
**Originality:** 3
**Overall Recommendation:** 4
**Confidence:** 4

**Summary:**

This paper proposes SyMerge, an extension of adaptive model merging methods that aims to move beyond non-interference toward task synergy. Building upon the AdaMerging framework, the method jointly optimizes merging coefficients and a single task-specific layer using an expert-guided self-labeling objective, instead of entropy minimization. Extensive experiments are conducted on vision classification, dense prediction, and NLP benchmarks. The results show that SyMerge consistently outperforms prior training-free and adaptive merging methods, particularly as the number of tasks increases.

**Compliance With Llm Reviewing Policy:**

Affirmed.

**Final Justification:**

Most of my worries have been resolved.

**Key Questions For Authors:**

The authors should address the weaknesses.

**Limitations:**

The authors should address the weaknesses.

**Strengths And Weaknesses:**

**Strength**

1. The paper proposes SyMerge, which builds upon the AdaMerging framework by introducing single-layer adaptation and a more stable self-labeling objective. This extension is technically simple yet practically effective, and represents a meaningful improvement over coefficient-only adaptive merging.

2. The method is evaluated on a wide range of tasks, including vision classification, dense prediction, and NLP benchmarks. Across these datasets, SyMerge consistently outperforms prior merging methods, demonstrating robustness and scalability as the number of tasks increases.

3. The paper includes experiments under corrupted inputs (distribution shift) and even merging models trained from different initializations (disjoint basins), where many existing merging approaches fail. These results highlight the practical stability and generalization ability of the proposed method.


**Weakness**

1.  Although SyMerge does not introduce additional parameters at inference time (same as Adamerging [1]), it updates a full task-specific layer during adaptation. In contrast, several baselines introduce lightweight additional modules (e.g., adapters) while keeping backbone layers frozen, resulting in substantially fewer trainable parameters. Consequently, the optimization flexibility across methods is not directly comparable. Moreover, Table I suggests that jointly optimizing both the merging coefficients and a task-specific layer yields significant performance gains, implying that the increased trainable capacity may be a key contributing factor. A comparison under matched trainable parameter budgets would help disentangle the effect of capacity from the proposed synergy mechanism and strengthen the fairness of the evaluation. Overall, SyMerge can largely be interpreted as AdaMerging augmented with single-layer training. It would therefore be important to clarify whether the observed gains arise primarily from the proposed synergy mechanism or from the additional trainable capacity introduced during adaptation.

2. While the paper argues that entropy-based optimization in AdaMerging [1] may lead to drift, it remains unclear why introducing single-layer adaptation would inherently prevent such drift or promote stable merging. The mechanism by which updating a task-specific layer facilitates representation alignment and improves cross-task compatibility is not theoretically justified. In particular, it is unclear why this additional flexibility would not further destabilize the merged encoder or overfit to pseudo-labels. A clearer analysis of how layer-level adaptation interacts with coefficient optimization would strengthen the claimed synergy mechanism. This is especially important under domain shift, where pseudo-label quality may degrade and additional trainable flexibility could potentially amplify errors rather than correct them.

3. From a conceptual standpoint, model merging is typically valued for being training-free [2,3]. By introducing layer-level updates, SyMerge departs from this principle and moves toward adaptive fine-tuning. While this extension is practically useful, it raises the question of whether the method should still be considered a pure merging approach or rather a form of lightweight multi-task adaptation built on top of merging.

4. To better isolate the contribution of layer-level adaptation, it would be informative to apply the same single-layer training procedure on top of strong training-free merging baselines (e.g., [2,3]). This would clarify how much of the performance gain is attributable to the proposed synergy mechanism versus the additional trainable flexibility introduced during adaptation.

[1] Yang, Enneng, et al. Adamerging: Adaptive model merging for multi-task learning, 2023

[2] Gargiulo, Antonio Andrea, et al. Task singular vectors: Reducing task interference in model merging, 2025

[3] Marczak, Daniel, et al. No task left behind: Isotropic model merging with common and task-specific subspaces, 2025

---

> ### Author Rebuttal · Authors · 2026-03-31
>
> We are grateful for the reviewer's valuable suggestions. We respond to each concern below.
>
> | **W1-1:** SyMerge uses more trainable parameters; gains may come from increased capacity|
> | :-
>
> We respectfully disagree with this concern. As shown in **Appendix Table I**, SyMerge is not simply a method that benefits from using more trainable parameters. For the **8-task ViT-B/32** setting, SyMerge (**90.1%**) uses only **0.39M** trainable parameters, significantly fewer than WEMoE (89.4%) which requires **7.16M** parameters. While Surgery (87.5%) uses **0.13M**, it employs a **2-stage training** (AdaMerging 500 + Surgery 1,000 iterations), making its total training cost higher than ours.
>
> Moreover, the merging coefficients add fewer than 0.001M parameters, so the capacity difference between layer-only and layer + coefficient settings is negligible. Yet the performance gap grows dramatically with more tasks (20 tasks: 75.0 → 88.6, Table 6). Training multiple layers, which directly increases capacity, consistently degrades performance (Figure 5). Thus, SyMerge's gains come from its design rather than parameter capacity.
>
> | **W1-2:** SyMerge can be viewed as AdaMerging + single-layer training; gains from synergy vs. capacity unclear|
> | :-
>
> SyMerge's core contribution is the finding that cross-task performance strongly predicts merge quality (Figure 2), and that adapting a single task-specific layer is sufficient to induce such synergy (Figure 3). This insight is not specific to AdaMerging: it also improves stronger training-free baselines such as TSV-M and ISO-Merging (**W4**).The two design choices, expert-guided self-labeling (Figure 4, Table F) and joint optimization of coefficients with the adapted layer (Table 6), follow naturally from this motivation. The gains instead arise from improved cross-task synergy (Figure 2, Table 4), not increased capacity (**W1-1**).
>
> | **W2-1:** Why single-layer adaptation prevents drift |
> | :-
>
> We apologize for the confusion. We do not claim that single-layer adaptation itself prevents drift; the stabilization comes from the self-labeling objective.
> Here, "drift" refers to the divergence of a proxy objective from supervised cross-entropy during training. As shown in Figure 4, entropy minimization suffers a sharp correlation drop with the ground-truth loss after training, whereas our objective leverages expert outputs and maintains high correlation throughout.
>
> | **W2-2:** Why single-layer adaptation improves cross-task compatibility |
> | :-
>
> In the merged model, each expert is overly specialized to its own task. Adapting a single layer with the expert signal makes the predictor more compatible with other tasks' encoders. This improves cross-task performance, which we empirically identify as a strong predictor of merge quality (Figures. 2-3) and is theoretically supported by Proposition 3.1.
>
> | **W2-3:** Why additional flexibility does not destabilize, and how the two components interact |
> | :-
>
> Figure 5 shows that training multiple layers degrades performance by disrupting task-agnostic knowledge, whereas a single layer preserves merging's generalization benefits. The resulting cross-task improvement ensures that adaptation and merging are mutually reinforcing rather than conflicting. This aligns with standard machine learning principles: fewer trainable parameters reduce overfitting risk.
>
> Finally, SyMerge was not designed to depend on coefficient optimization, and the adapted layer already improves performance when combined with fixed merging such as Task Arithmetic (Table 4). The observed complementarity likely arises because only a single layer is adapted while the coefficients handle merging across all layers. We plan to study this interaction further and will note it in the limitation section.
>
> | **W3:**  SyMerge departs from the training-free principle of model merging toward adaptive fine-tuning |
> | :-
>
> We position SyMerge alongside adaptive test-time merging methods such as AdaMerging, Surgery, and WEMoE, motivated by the weakness of training-free methods under distribution shift (Figure 1). SyMerge still remains within the **model merging** paradigm because it preserves the **task vector-based merging structure** and adapts it **without labeled data or multi-task supervised training**.
>
>
> | **W4:** Single-layer training should be applied on top of strong training-free baselines to isolate its contribution  |
> | :-
>
> Following the reviewer's suggestion, we applied the same single-layer adaptation to the frozen merged encoders of  **TSV-M** and **ISO-Merging** in the 20-task ViT-B/32 setting. TSV-M improves from 76.6% → **87.3%**, and ISO-Merging from 76.9% → **87.4%**. However, SyMerge still achieves the highest performance at **88.6%**, despite starting from a weaker Task Arithmetic initialization (**60.8%**). These results suggest the gains are not fully due to trainable flexibility alone, but to the joint optimization of merging coefficients and task-specific layers.

---

> > ### Author Rebuttal · Reviewer_MNZH · 2026-04-02
> >
> > Dear Authors,
> >
> > Thanks for the rebuttals. Your responses have addressed most of my concerns, and I appreciate the clarifications you provided. Based on this, I am comfortable increasing my score to 4.
> >
> > Best regards,
> >
> > Reviewer MNZH

---

> > > ### Author Response · Authors · 2026-04-03
> > >
> > > Thank you for the valuable feedback and for carefully considering our rebuttal. We are glad that our responses addressed most of your concerns. If you have any follow-up questions, we would be happy to address them.

---

### Official Review · Reviewer_Gbff · 2026-03-04

**Soundness:** 3
**Presentation:** 3
**Significance:** 3
**Originality:** 3
**Overall Recommendation:** 5
**Confidence:** 4

**Summary:**

The paper studies model merging, which aims to combine independently fine-tuned models into a single multi-task model without joint training. The research’s central contribution is to reframe the objective of model merging from merely avoiding task interference to actively enabling task synergy, in which tasks improve one another through better representation alignment. The authors introduce SyMerge, a lightweight test-time adaptive method that jointly optimizes encoder merging coefficients and a single task-specific layer using unlabeled data. To provide stable supervision, the method uses an expert-guided self-labeling objective based on predictions from individually fine-tuned models, rather than entropy minimization. Experiments across vision classification, dense prediction, and NLP benchmarks demonstrate that SyMerge consistently improves multi-task performance over existing merging approaches and remains effective even when merging models trained from different initializations.

**Compliance With Llm Reviewing Policy:**

Affirmed.

**Final Justification:**

This paper proposes SyMerge, a model-merging method that explicitly encourages task synergy rather than merely avoiding interference. The approach is technically sound, well-motivated, and supported by strong empirical results across diverse tasks. The formulation is clear, and the combination of test-time adaptation with expert-guided self-labeling is a compelling and practical contribution.

In my initial review, my main concern was the lack of a direct evaluation of task synergy, particularly at the per-sample level. The rebuttal addressed this convincingly by providing a detailed analysis of positive and negative transfer, which directly supports the paper’s central claim. Additional clarifications on the method and experimental setup further improved the paper’s clarity.

Overall, the rebuttal significantly increased my confidence in the work. I consider this a meaningful contribution to the model merging literature and recommend acceptance.

**Key Questions For Authors:**

### Questions
1) *Interpretation of merging coefficient in Figure 3b.*
Figure 3b shows a strong dependence of the cross-task performance gains on the merging coefficient. The paper interprets this as evidence for “the importance of a well-formed merged encoder.” Could the authors clarify why a lower merging coefficient corresponds to a better-formed merged encoder in this context?
2) *Direct measurement of task synergy.* The paper motivates SyMerge as enabling task synergy, but the experiments mainly report aggregate task performance. Have the authors considered directly measuring positive transfer between tasks?
3) *Clarification of the SyMerge objective and relation to distillation.* In Equation (3), the method minimizes cross-entropy between the merged model predictions and the expert model outputs. Are these hard pseudo-labels or soft probability distributions from the expert models? If soft targets are used, this setup appears closely related to knowledge distillation. Could the authors clarify this aspect and discuss how the objective differs from or relates to standard distillation approaches?

**Limitations:**

While the paper briefly notes a methodological limitation in its reliance on the quality of the expert models used for self-labeling, other relevant limitations are not discussed. In particular, the method requires access to an unlabeled merge set for test-time adaptation, which may not always be available in practice (see weaknesses). The authors could strengthen the paper by briefly discussing this practical limitation and clarifying in which settings such data would realistically be accessible. Additionally, although the work is largely methodological, a brief statement on potential risks of merging models trained on different data sources (e.g., unintended bias transfer or domain mismatch) would further strengthen the limitations section.

**Strengths And Weaknesses:**

### Strengths

**Soundness**:
The paper is technically solid and supports its claims through both theoretical and empirical analysis. The authors motivate their approach with pilot studies that examine the relationship between cross-task performance and merge quality. A theoretical analysis further connects improvements in cross-task performance to improved merge performance under reasonable assumptions. The empirical evaluation is extensive, covering vision classification, dense prediction, and NLP tasks, and compares against a large number of strong baselines. Additional ablation studies analyzing the role of merging coefficients and task-specific layer adaptation further strengthen the empirical evidence.

**Presentation**:
The paper is clearly written and well structured. The progression from identifying limitations of existing merging methods to motivating the concept of task synergy and introducing SyMerge is easy to follow. The pilot studies help build intuition for the method, and figures and tables effectively support the explanations and results.

**Significance**:
The paper addresses an important problem in model merging and multi-task learning. Moving beyond avoiding task interference toward actively leveraging task synergy provides a meaningful conceptual shift that could influence future work. The strong empirical results across diverse tasks also suggest that the method may have practical utility.

**Originality**:
The paper introduces a novel perspective on model merging by framing the objective in terms of task synergy rather than non-interference. SyMerge combines test-time adaptation, self-labeling, and parameter merging in a simple framework that jointly optimizes merging coefficients and a single task-specific layer, resulting in a method that is both conceptually interesting and practically lightweight.

### Weaknesses

**Soundness**:
The paper’s central motivation is that SyMerge enables task synergy, where knowledge from other tasks improves performance on a given task. However, the paper does not explicitly evaluate or quantify such positive transfer. A more direct analysis could be conducted at the per-sample prediction level. For a given task B, the authors could compare the merged model's predictions with those of the corresponding expert model trained on task B. In particular, the fraction of samples for which the expert model predicts incorrectly while the merged model predicts correctly would provide a direct estimate of positive transfer from other tasks. If SyMerge indeed leverages cross-task synergies, this quantity should be higher than that of merging baselines that primarily optimize for non-interference. Including such an analysis would be essential to validating the paper’s core claim.

Some aspects of the experimental setup and method formulation also lack clarity. In the pilot study presented in Section 3 and Figure 3, it is unclear which merging strategy is used to construct the merged encoder and how many models are involved in the merge. It is further unclear how many models are merged here. Additionally, the SyMerge objective is described as minimizing cross-entropy between merged model predictions and expert outputs, but it is unclear whether this uses hard pseudo-labels or soft probability distributions. This setup appears closely related to knowledge distillation, yet the paper does not discuss this relation.

**Presentation**:
Although the paper is clearly written overall, several experimental details could be clarified. In the experiments shown in Figure 5, the paper claims that adapting a single internal layer achieves performance comparable to adapting the classifier, but it is unclear which specific layer is used in this configuration and whether the observation holds consistently across layers. Providing this information would make the analysis easier to interpret and reproduce.

The paper reports the performance of the individual task models as an “upper bound” in several tables. However, under the paper’s central motivation of exploiting task synergy, the individual model performance should not strictly serve as a true upper bound, since positive transfer between tasks could, in principle, enable the merged model to surpass individual models. This framing somewhat contradicts the conceptual motivation of the work.

**Significance**:
One practical limitation that is not discussed in Section 5 is the reliance on a merge set for test-time adaptation. While this requirement is common for adaptive merging methods, it can be restrictive in settings where suitable data is unavailable or proprietary, a common issue in LLM research. A brief discussion of this limitation would strengthen the paper.

**Originality**:
The paper introduces the notion of explicitly encouraging task synergy during model merging. However, the concept of positive forward transfer between tasks has been extensively studied in the continual learning literature. The paper would benefit from discussing this connection more explicitly and positioning its contribution relative to prior work on positive transfer and task synergy in multi-task or continual learning settings.

---

> ### Author Rebuttal · Authors · 2026-03-31
>
> We appreciate the reviewer's detailed and insightful comments. Our responses are as follows.
>
> | **W1 & Q2:**  Positive transfer is not directly measured or quantified at the per-sample level|
> | :-
>
> We thank the reviewer for this suggestion. This has already been analyzed at the per-sample prediction level in Appendix Figure G. The table below summarizes the results, reporting (1) negative transfer (expert correct / merged wrong), (2) positive transfer (expert wrong / merged correct), and the net difference:
>
> | Metric | TA | Ties | AdaMerging | Surgery | EMR | Ours |
> | :--- | :--- | :--- | :--- | :--- | :--- | :--- |
> | **(1) NT** ↓ | 18790 | 15677 | 9608 | 6240 | 3022 | 2408 |
> | **(2) PT** ↑ | 1571 | 1865 | 1964 | 1845 | 1432 | 1593 |
> | **NT-PT** ↓ | 17219 | 13812 | 7644 | 4395 | 1590 | 815 |
>
> SyMerge achieves the smallest net difference, demonstrating that it best preserves individual model knowledge while benefiting from merging. We note that examining positive transfer alone is insufficient to capture synergy, as gains on new samples may come at the cost of losing previously correct predictions. We will move this analysis to the main paper and more explicitly report both directions as suggested.
>
> | **W2:**  Pilot study details unclear (merging strategy, number of merged models)|
> | :-
>
> We apologize for the confusion. In the Figure 3 pilot study, the merged encoder is constructed by **merging 8 task models with Task Arithmetic**, with merging coefficients ranging from 0.1 to 1.0. These details are currently provided in Appendix A.1, and we will make them explicit in the main paper.
>
> | **W3 & Q3:**  Hard vs. soft pseudo-labels unclear; relation to knowledge distillation not discussed|
> | :-
>
> Our main experiments are conducted using **hard pseudo-labels**, and a comparison using **soft probability distributions** (via KL divergence) is already provided in Appendix Figure C, where both variants yield nearly identical performance. While expert outputs are used as supervision, SyMerge is distinct from knowledge distillation. It does not train a separate student model, but jointly optimizes merging coefficients and a task-specific layer within the merged model for cross-task alignment. We will clarify this distinction in the final version.
>
> | **W4:**  Which specific layer is used in Figure 5, and whether the observation is consistent across layers|
> | :-
>
> Since CLIP ViT-B/32 consists of **12 transformer layers**, Figure 5 shows results when each individual layer is trained together with the merging coefficients. For example, a value of 3 on the x-axis of Figure 5 indicates training the 3rd transformer layer. Additionally, in the legend, **"early layers"** refers to layers 1–6, and **"late layers"** refers to layers 7–12. As shown in Table E, the comparable performance holds consistently across all single layers. We will clarify this in the main paper.
>
> | Layer | Single layer (1~12) | Late | Early | All |
> | :--- | :--- | :--- | :--- | :--- |
> | **Avg.** | 90.0 ~ 90.5 | 88.4 | 89.4 | 86.2 |
>
>
> | **W5:** If synergy enables positive transfer, individual model performance should not be treated as an upper bound, which contradicts the paper's motivation|
> | :-
>
> Although we adopted the term "upper bound" from **prior model merging literature**, we agree that the term "upper bound" is not the most accurate description in our setting, as positive transfer can allow the merged model to exceed individual model performance on some tasks. We will revise this wording and instead present individual model performance as a single-task reference point.
>
> | **W6:** Reliance on a merge set for test-time adaptation is not discussed as a limitation|
> | :-
>
> We thank the reviewer for the suggestion. We will add this point to the **limitation section**.
>
> | **W7:** Connection to positive forward transfer in continual learning literature not discussed |
> | :-
>
> We thank the reviewer for the suggestion. We will include the relevant references as **related work in the appendix**.
>
> | **Q1:** Why lower merging coefficients yield better cross-task performance in Figure 3b |
> | :-
>
> We apologize for the confusion. **Figure 3b** is not intended to argue that lower merging coefficients always produce better encoders. Rather, it shows that **cross-task alignment improvement is sensitive to the merging coefficient choice**, as also discussed in Appendix B.2.
>
> In Task Arithmetic, larger coefficients cause the merged encoder to more strongly reflect each task vector's direction. Classifiers trained on such an encoder may become overly specialized to that particular merged representation, reducing cross-task compatibility when paired with other individual encoders. This explains why smaller coefficients tend to yield higher cross-task performance in the pilot study.
>
> | **L1:**  Other relevant limitations beyond expert model quality are not discussed|
> | :-
>
> We thank the reviewer for the suggestion and will add this point to the **limitation section**.

---

> > ### Author Rebuttal · Reviewer_Gbff · 2026-04-01
> >
> > Thank you for the clear and thorough rebuttal. I appreciate the additional analyses and detailed clarifications, which address my main concerns. In particular, the provided per-sample analysis of positive and negative transfer is very helpful and directly strengthens the paper’s central claim about task synergy. Overall, I am satisfied with the revisions and will increase my score accordingly.

---

> > > ### Author Response · Authors · 2026-04-03
> > >
> > > Thank you sincerely for the thoughtful review and for evaluating our rebuttal so thoroughly. We greatly appreciate your detailed feedback, especially regarding the direct measurement of task synergy, which has helped us strengthen the paper. We will incorporate the discussed revisions into the final version.

---

### Official Review · Reviewer_uvfn · 2026-03-10

**Soundness:** 3
**Presentation:** 3
**Significance:** 3
**Originality:** 2
**Overall Recommendation:** 4
**Confidence:** 3

**Summary:**

This paper proposes a lightweight test-time adaptive framework SyMerge, which shifts the objective of model merging from avoiding task interference to improve task synergy. The method achieves this by jointly optimizing merging coefficients and a single task-specific layer using expert-guided self-labeling. The approach is evaluated across a wide range of vision and NLP tasks, achieving state-of-the-art results.

**Compliance With Llm Reviewing Policy:**

Affirmed.

**Final Justification:**

The authors' rebuttal addressed my main concerns regarding the robustness to poor expert models and the computational overhead of the expert supervision. The paper's motivation is insightful, and the methodology is sound. Since the response fully resolved my questions, I will maintain my original score of Weak Accept.

**Key Questions For Authors:**

1. SyMerge relies on expert models to generate self-labels. If the experts vary significantly in quality, or if some of them perform poorly, would this negatively affect the performance of the merged model?

2. Could the authors provide the total runtime overhead during the adaptation stage and discuss how it scales with the number of experts $K$?

3. During the generation of self-labels, is the inference overhead of the expert models included in the Peak GPU Memory reported in Appendix Table I?

**Limitations:**

yes

**Strengths And Weaknesses:**

Strengths

1. The motivation is clear. Shifting the goal of model merging from ensuring non-interference to achieving positive synergy is an insightful idea. The paper is also well written and easy to follow.

2. The proposed model merging method is simple yet effective. It performs well across a variety of vision and NLP tasks, and comprehensive ablation studies further demonstrate its effectiveness and robustness.

Weaknesses

1. The method depends on the predictions of individual expert models to generate self-labels. While the authors briefly acknowledge this in the Limitations section, the paper currently lacks a ablation study. It remains unclear how robust the merging process is if a subset of the expert models exhibits severe bias, corruption, or exceptionally poor performance.

2. The implementation details regarding the expert supervision are somewhat vague. It is not specified whether the predictions from the individual expert models are pre-computed and cached before adaptation or generated on-the-fly during training. Since the method is described as lightweight, clarifying this aspect is important for understanding the practical computational cost. If generated on-the-fly, this would introduce a non-trivial computational and time overhead proportional to the number of tasks ($K$).

---

> ### Author Rebuttal · Authors · 2026-03-31
>
> We thank the reviewer for the valuable feedback. We address each point below.
>
> | **W1 & Q1:**  Robustness to low-quality or biased expert models is unclear |
> | :-
>
> While **Figure 6a** shows performance variation across supervisory experts, it does not directly address severe bias or corruption scenarios. To investigate this, we conducted two additional stress tests.
>
> ### (1) Random-Corruption Stress Test
>
> In the 8-task setting, we replaced a subset of experts with **Gaussian random-weight experts** matched in mean and variance. As shown in the table below, when only a single expert is severely corrupted, SyMerge experiences an overall accuracy drop (90.1 → 77.3), but the degradation is **primarily confined to the corrupted task**, with performance on the remaining tasks largely preserved. In contrast, other adaptive baselines suffer **near-complete collapse across all tasks** even when only a single expert is corrupted. This suggests that SyMerge can **limit the negative impact** of a small number of corrupted experts. However, when two or more experts are corrupted, SyMerge also degrades significantly, and we do not claim that our method is fully robust to arbitrary levels of severe expert corruption.
>
> | Method | Clean Avg. ↑ | Corrupted Avg. ↑ |
> | :--- | ---: | ---: |
> | TA | 69.1 | 6.0 |
> | AdaMerging | 80.1 | 4.6 |
> | Surgery | 87.5 | 18.8 |
> | WEMoE | 89.4 | 4.7 |
> | SyMerge | **90.1** | **77.3** |
>
> ### (2) Backdoored-Expert Stress Test under the BadMerging Setup
>
> We further experimented by replacing one out of 6 experts (CIFAR100) with a **backdoored expert** following **BadMerging** [1]: a model that behaves normally on clean inputs, but produces attacker-specified outputs when a specific trigger patch is present. We specifically report the **off-task backdoor** setting, which measures whether a backdoored CIFAR100 expert, once merged, can transfer its trigger effect to other tasks (e.g., Cars) — a more diagnostic evaluation of cross-task vulnerability.
>
> As shown in the table below, SyMerge achieves the **highest Clean Accuracy and Backdoored Accuracy** (accuracy when merging only clean experts, and accuracy when one backdoored expert is included in the merge, respectively), while simultaneously exhibiting the **lowest Off-task Attack Success Rate** among all adaptive merging methods. This demonstrates that SyMerge remains relatively more stable than other adaptive merging approaches even in the presence of intentionally poisoned experts. We will incorporate these experimental results into the final version of the paper.
>
> | Method | Clean ↑ | Backdoor ↑ | Off-task ASR ↓ |
> | :--- | ---: | ---: | ---: |
> | TA | 76.3 | 76.0 | 95.6 |
> | TIES | 74.5 | 74.4 | 90.6 |
> | RegMean | 77.7 | 77.5 | 90.0 |
> | AdaMerging | 83.1 | 82.9 | 98.3 |
> | Surgery | 84.5 | 84.6 | 86.6 |
> | SyMerge | **86.1** | **86.0** | **75.6** |
>
> `Reference`
> [1] *BadMerging: Backdoor Attacks against Model Merging, CCS 2024*
>
> | **W2 & Q2:** Expert supervision details unclear (pre-computed vs. on-the-fly), and potential computational overhead |
> | :-
>
> In our current implementation, expert predictions are generated **on-the-fly** during adaptation. Only the single expert corresponding to the current task is invoked at each step, so the per-step overhead does not scale by a factor of K. Caching is also feasible, but in the **online test-time adaptation/streaming setting** where new test samples arrive sequentially, on-the-fly supervision is more natural
>
> To clarify the actual cost, we measured per-step runtime, with results shown in the table below. SyMerge's on-the-fly implementation is moderately slower than AdaMerging, but is comparable to Surgery and substantially faster than WEMoE. Furthermore, enabling expert prediction caching reduces the per-step cost to **111ms (8-task)** and **195ms (20-task)**, bringing it to nearly the same level as AdaMerging.
>
> While expert supervision does introduce additional computation, the overhead is **manageable** and can be largely mitigated via the cached variant. We believe this represents a reasonable trade-off given the consistent performance improvements observed across all benchmarks (e.g., AdaMerging 69.6% vs. SyMerge **88.6%** on 20-task ViT-B/32). We will incorporate these implementation details and runtime analysis into the final version of the paper.
>
> | # Tasks | AdaMerging (ms) | Surgery (ms) | WEMoE (ms) | Ours (ms) | Ours-Cached (ms) |
> | :---: | ---: | ---: | ---: | ---: | ---: |
> | 2 | 71 | 170 | 331 | 170 | 65 |
> | 4 | 87 | 183 | 331 | 179 | 80 |
> | 8 | 114 | 211 | 428 | 215 | 111 |
> | 14 | 162 | 259 | 743 | 262 | 153 |
> | 20 | 207 | 299 | 1047 | 303 | 195 |
>
>
> | **Q3:** Whether expert model inference overhead is included in Table I's peak GPU memory |
> | :-
>
> Yes. The reported peak GPU memory includes expert inference under the same sequential-update setting. In our implementation, only one expert is temporarily loaded onto the GPU at a time, rather than keeping all experts in memory simultaneously.

---

> > ### Author Rebuttal · Reviewer_uvfn · 2026-04-02
> >
> > Thanks for the author's response. Most of my concerns are addressed. I will keep my original score.

---

> > > ### Author Response · Authors · 2026-04-03
> > >
> > > Thank you for the follow-up and for taking the time to read our rebuttal. We are glad that our response was helpful. If any further clarification would be useful during the discussion period, we would be happy to provide it.

---

### Official Review · Reviewer_jMhd · 2026-03-13

**Soundness:** 3
**Presentation:** 3
**Significance:** 3
**Originality:** 3
**Overall Recommendation:** 5
**Confidence:** 3

**Summary:**

This paper proposes SyMerge, a gradient-based model merging framework capable of merging models trained on settings beyond classification, and is robust to models with different initializations. SyMerge does this by minimizing output "distance" (where distance is defined according to the setting -- e.g., l1 loss for regression tasks) between a merged model and the base model for each underlying task.
It further optimizes the task-specific layers (in addition to scaling coefficients) under the objective, using unlabeled data. SyMerge outperforms baselines in nearly all settings, and contains extensive analysis.

**Compliance With Llm Reviewing Policy:**

Affirmed.

**Final Justification:**

The authors have resolved all my questions.

**Key Questions For Authors:**

My three points in the weaknesses section are my questions.

**Limitations:**

Yes

**Strengths And Weaknesses:**

**Strengths:**
- SyMerge substantially outperforms prior work in across a diverse array of settings, extending beyond the predominant classification benchmarks in this field. It is robust to merging models of different initializations too.
- SyMerge has roughly the same computational expense as other gradient-based approaches, while outperforming them
- The method is fairly well motivated (see comment in weaknesses) with a decent pilot study.

**Weaknesses:**

1) Please correct me if I'm wrong, I don't understand the utility of using cross-task alignment as a motivation for the method. The paper is primarily interested in enhancing model merging performance, and proposes a method that aims to align the merged model's outputs with the outputs of each underlying base model, such that only the scaling coefficients and the task-specific parameters of the merged model are updated. This is great, especially because the paper's results are quite strong. However, the paper's method does not explicitly optimize for better cross-task alignment. That is, rather than going from improving cross-task alignment to improving merging performance, the authors seem to do the opposite.
- I don't think cross-task alignment is needed as a method motivation. The authors have done a good pilot study, and Figure 2 is interesting, but I don't see how it relates to their approach. Perhaps, the authors can keep it disjoint from their method?
- Building on these, I'm not convinced that the experiment conducted in the Section 3 intro is indicative that improving cross-task alignment leads to better merging performance. In the experiment, the merged encoder is fixed ahead of time, so I have trouble seeing how improving cross-task alignment improves performance from this example.
2) The ablation on not using the task-specific layer for training is not quite right (the Improvement beyond classifier adaptation paragraph). To conduct this properly, the authors should freeze the task-specific layers and only train the scaling coefficients with their objective function, then evaluate the performance. Currently there is a confounding variable because the merged model was obtained with both optimizing the scaling coefficients and task-specific layers. Table 6 row 1 and Table D row 3 should be recomputed.
3) Do all models have to be stored in memory during training?

---

> ### Author Rebuttal · Authors · 2026-03-31
>
> We thank the reviewer for the constructive feedback. We address each point below.
>
> | **W1-1:** Cross-task alignment not explicitly optimized despite being used as motivation. |
> | :-
>
>
> We appreciate the reviewer's thoughtful comment. The cross-task alignment analysis is not an optimization objective but a **structural motivation** for why single-layer adaptation helps merging. Our reasoning is as follows:
>
> 1. **Figure 2** empirically demonstrates a strong correlation between cross-task alignment and merge performance.
> 2. The **pilot study** in Section 3 is a controlled experiment designed to test whether single-layer adaptation can directly improve cross-task functional compatibility. Its role is to isolate this mechanism, rather than to establish the full benefit of the final method.
> 3. **Section 3.2** provides a theoretical justification for why improving cross-task performance leads to better merge performance.
>
> SyMerge is trained with an **expert-guided self-label cross-entropy objective**. While cross-task alignment is not directly optimized, it provides a principled explanation for why this objective improves merging. We will revise the presentation to make this logical structure clearer.
>
> | **W1-2:**  Cross-task alignment analysis seems disjoint from the actual method |
> | :-
>
> We believe this concern stems from a **presentation issue rather than a flaw in the underlying logic**. The cross-task alignment analysis does not stand as an independent observation; rather, it plays a central role in connecting two key questions:
> *"Why does task-specific layer adaptation help?"* and *"Why does it lead to better merge performance?"*
>
> Without the cross-task alignment analysis (Figure 2 + Section 3.2), the pilot study would lack a principled explanation for **why learning task-specific layers benefits the merged model**. Each component thus plays a distinct and necessary role in the overall narrative. We will revise the paper to more clearly delineate the role of each component.
>
> | **W1-3:**  Pilot study with frozen encoder does not link cross-task alignment to merge performance |
> | :-
>
> The pilot study is a controlled experiment with a frozen encoder, designed to isolate that task-specific layer adaptation alone can substantially improve cross-task performance. As the reviewer notes, this alone does not establish that improved alignment leads to better merging. That connection is separately established by Figure 2 (empirical correlation) and Section 3.2 (theoretical justification). The pilot study answers how cross-task alignment can be improved in practice, and Table 4 confirms this extends beyond the controlled setting: classifiers learned by SyMerge improve both merged-task and cross-task performance when transferred to different merged encoders. We will revise the presentation to make this logical structure clearer.
>
> | **W2:**  Ablation should use coefficient-only training with frozen task-specific layers. |
> | :-
>
> We apologize for the confusion. The setting requested by the reviewer was already included in **Table 6, Row 1**, where the task-specific layers are frozen and only the merging coefficients are optimized via the SyMerge objective.
>
> We believe the reviewer's confusion arose because the intent of the **"Improvement beyond classifier adaptation"** paragraph was not stated clearly enough. This paragraph serves a different purpose: it analyzes whether SyMerge's joint optimization improves the merged encoder itself. We retain SyMerge's encoder but replace the fine-tuned classifier with the original zero-shot classifier at evaluation. The substantial gain over Task Arithmetic (which uses the same zero-shot classifier) confirms that the encoder quality itself has improved, independent of classifier adaptation.
>
> In summary, Table 6 Row 1 is a coefficient-only ablation, while this paragraph is an encoder quality analysis. We will revise the wording to clearly distinguish these two experiments.
>
>
> | **W3:**  Do all models have to be stored in memory during training? |
> | :-
>
> SyMerge does not require all expert models in GPU memory simultaneously. As in Surgery, expert models are kept on CPU and only the single expert needed for pseudo-label generation is temporarily moved to GPU, following the sequential per-task update strategy detailed in Appendix A.
> As shown in Table I, SyMerge's peak GPU memory (7.12GB on ViT-B/32, 23.71GB on ViT-L/14) is comparable to AdaMerging (7.09GB, 22.88GB) and substantially lower than WEMoE (8.01GB, 33.57GB). The trainable parameters are limited to a single task-specific layer and merging coefficients (0.39M on ViT-B/32), and memory does not scale with the number of experts.

---

> > ### Author Rebuttal · Reviewer_jMhd · 2026-04-04
> >
> > The authors have resolved my concerns, and I've adjusted my score accordingly.

---

> > > ### Author Response · Authors · 2026-04-06
> > >
> > > Thank you for your time and consideration throughout the review process. We appreciate your engagement with our rebuttal and are pleased that our clarifications were helpful. We will reflect the discussed changes in the final version.

---

### Decision · Program_Chairs · 2026-04-30

**Decision:**

Accept (regular)

**Comment:**

This paper works towards the model merging in multi-task learning by shifting the primary objective from simply avoiding task interference to actively enabling task synergy. Based on this, the authors introduce a lightweight test-time adaptive framework named SyMerge, which jointly optimizes merging coefficients alongside a single task-specific layer. To provide stable supervision, SyMerge utilizes an expert-guided self-labeling objective rather than relying on entropy minimization. The method is evaluated across vision classification, dense prediction, and NLP benchmarks, achieving state-of-the-art results and demonstrating robustness even when merging models trained from different initializations.

During the rebuttal, the authors systematically and convincingly addressed all reviewer concerns, which led to increased confidence and raised scores from the reviewers:
- Validation of Task Synergy: Reviewer Gbff initially pointed out that the paper lacked a direct, per-sample evaluation of positive transfer to validate the claim of "task synergy". In response, the authors provided a detailed per-sample analysis of positive and negative transfer, demonstrating that SyMerge achieves the best net difference (positive transfer minus negative transfer) compared to baselines. This directly strengthened the paper's central claim and satisfied the reviewer.
- Parameter Capacity vs. Synergy: Reviewer MNZH raised a concern that the performance gains might simply stem from the increased parameter capacity of training a full task-specific layer compared to coefficient-only baselines. The authors countered this by showing that SyMerge uses significantly fewer trainable parameters (0.39M) than other adaptive baselines like WEMOE (7.16M) while achieving better performance. Furthermore, they demonstrated that adapting multiple layers actually degrades performance, proving that the gains come from the synergistic design rather than sheer capacity.
- Robustness to Corrupted Experts: Reviewer uvfn inquired about the method's robustness if the expert models used for self-labeling are biased or corrupted. The authors provided two additional stress tests: a random-corruption test and a backdoored-expert test. These tests showed that SyMerge confines the degradation primarily to the corrupted task and exhibits the lowest off-task attack success rate among adaptive methods.
- Computational Overhead: In response to queries regarding the computational cost of on-the-fly expert supervision, the authors provided a runtime analysis showing that SyMerge's per-step cost is manageable and can be further reduced to match AdaMerging by enabling expert prediction caching.

In summary, this paper presents an original and well-motivated perspective on model merging. The methodology is technically sound, practical, and comprehensively evaluated across varied domains. The authors are expected to incorporate all the clarifications, the per-sample positive transfer analysis, and the newly discussed limitations into the camera-ready version, as promised during the rebuttal phase.